# High-Throughput Synthesis of Nanogap-Rich Gold Nanoshells Using Dual-Channel Infusion System

**DOI:** 10.3390/ijms25031649

**Published:** 2024-01-29

**Authors:** Yoon-Hee Kim, Hye-Seong Cho, Kwanghee Yoo, Kyeong-Min Ham, Homan Kang, Xuan-Hung Pham, Bong-Hyun Jun

**Affiliations:** 1Department of Bioscience and Biotechnology, Konkuk University, Seoul 05029, Republic of Korea; yoonhees@konkuk.ac.kr (Y.-H.K.); joh0302@konkuk.ac.kr (H.-S.C.); heu1997@konkuk.ac.kr (K.Y.); hkm7321@konkuk.ac.kr (K.-M.H.); 2Gordon Center for Medical Imaging, Department of Radiology, Massachusetts General Hospital and Harvard Medical School, Boston, MA 02114, USA; hkang7@mgh.harvard.edu

**Keywords:** gold nanoshell, high-throughput synthesis, seed-mediated growth, dual-channel infusion, localized surface plasmon resonance, surface-enhanced Raman scattering

## Abstract

Gold nanoshells have been actively applied in industries beyond the research stage because of their unique optical properties. Although numerous methods have been reported for gold nanoshell synthesis, the labor-intensive and time-consuming production process is an issue that must be overcome to meet industrial demands. To resolve this, we report a high-throughput synthesis method for nanogap-rich gold nanoshells based on a core silica support (denoted as SiO_2_@Au NS), affording a 50-fold increase in scale by combining it with a dual-channel infusion pump system. By continuously dropping the reactant solution through the pump, nanoshells with closely packed Au nanoparticles were prepared without interparticle aggregation. The thickness of the gold nanoshells was precisely controlled at 2.3–17.2 nm by regulating the volume of the reactant solution added dropwise. Depending on the shell thickness, the plasmonic characteristics of SiO_2_@Au NS prepared by the proposed method could be tuned. Moreover, SiO_2_@Au NS exhibited surface-enhanced Raman scattering activity comparable to that of gold nanoshells prepared by a previously reported low-throughput method at the same reactant ratio. The results indicate that the proposed high-throughput synthesis method involving the use of a dual-channel infusion system will contribute to improving the productivity of SiO_2_@Au NS with tunable plasmonic characteristics.

## 1. Introduction

Gold nanoshells are structures in which gold layers are stacked at the nanometer scale; these have been actively studied in various fields because of their unique plasmonic properties [1,2,3,4]. Compared to individual gold nanoparticles (AuNPs), gold nanoshells exhibit a wide range of localized surface plasmon resonance (LSPR) frequencies due to plasmon hybridization between the dielectric core and metal shell [5]. Because of this feature, gold nanoshells can be used as optical antennas that respond to light sources of various wavelengths [6]. Leveraging their optical properties, gold nanoshell composites have been widely applied in various fields such as photocatalysis, selective drug delivery, and plasmonic sensing [7,8,9].

As a major application, gold nanoshells can be utilized as a surface-enhanced Raman spectroscopy (SERS) substrate, which enhances the Raman scattering signal of molecules adsorbed on the nanostructured surface [10]. Gold nanoshells can adsorb thiol-containing Raman labels with high affinity and are suitable for long-term storage due to their high structural stability [11]. In addition, near-infrared (NIR) lasers can be selected for SERS analysis, corresponding to the LSPR frequency of gold nanoshells ranging from the visible to NIR regions [12]. The excellent skin penetration at these laser wavelengths allows gold nanoshells to avoid autofluorescence from the body, enabling the in vivo tracking of target substances inside cellular tissues [13].

The plasmonic characteristics of gold nanoshells, including SERS, are closely related to the shell thickness and surface nanostructures [14,15]. Recently, we reported a method for preparing gold nanoshells on silica nanospheres (denoted as SiO_2_@Au NS) with abundant nanogaps between the closely packed AuNPs [16]. The thickness of the gold shell in SiO_2_@Au NS was adjusted to the order of tens of nanometers by controlling the Au precursor concentration during the growth process. Because of its nanogap-rich surface, which can act as an electromagnetic hotspot, SiO_2_@Au NS generated strong SERS signals with an enhancement factor of approximately 3.8 × 10^6^ [17]. Given these advantages, the seed-mediated growth approach is useful for tuning the plasmonic characteristics, which depend on the thickness of the gold shell. However, the methodology for preparing SiO_2_@Au NS involves the repetitive addition of each precursor solution at 5 min intervals for the growth of seeded AuNPs, which is a time-consuming and labor-intensive process with a limited amount of the synthesized material. As alternative methods for improving the throughput of gold nanoshell synthesis, several microfluidic systems using injection pumps have been reported. However, these methods require simultaneous pumping of oil and gas in addition to each reactant solution to achieve multi-phase microfluidic or require a specially designed microreactor [18,19].

Herein, we report a novel high-throughput method for SiO_2_@Au NS synthesis on a simple infusion system with a newly optimized seed-mediated growth process at a 50-fold higher scale than that achieved using our previously reported method [16]. To obtain a homogeneous, closely packed AuNP surface in a high-throughput environment, both precursor solutions (aqueous Au^3+^ and ascorbic acid) were added dropwise at a constant flow rate using a dual-channel syringe pump during the seed-mediated growth of AuNP-seeded silica NPs. The thickness control of the gold shell achieved using the previously reported low-throughput method was successfully reproduced even on the 10 mg scale of the proposed high-throughput synthesis method, and a linear relationship was found between LSPR and SERS signals. As a SERS substrate, the high performance of SiO_2_@Au NS prepared by the proposed method was confirmed on the basis of the signal of the adsorbed Raman labels.

## 2. Results and Discussion

### 2.1. High-Throughput Synthesis of SiO_2_@Au NS

High-throughput method for the synthesis of SiO_2_@Au NS proposed in this work is based on a previously reported seed-mediated growth approach [16,20]. As shown in Figure 1a, the method proceeded in two steps: AuNP seed immobilization and seed-mediated growth. First, AuNP seeds were immobilized on silica NPs aminated by a sol–gel reaction with (3-aminopropyl)trimethoxysilane (APTS). Then, AuNP seeds immobilized on the silica surface were grown by the simultaneous addition of a gold precursor solution and an ascorbic acid solution. For AuNP-seed-mediated growth of SiO_2_@Au NS prepared at a scale of 10 mg, we constructed a continuous infusion system using a dual-channel syringe pump, as shown in Figure 1b.

The syringes containing the Au^3+^ precursor solution and ascorbic acid (AA) solution were fixed to each pump and connected to the reaction mixture through the silicone tubings. AuNP-seed silica NPs dispersed in aqueous polyvinylpyrrolidone (PVP) solution were placed in a glass vial and gently mixed on the magnetic stirrer. By continuously adding each solution to the reaction solution at a fixed flow rate of 36.67 µL/min, we attempted to control the growth of the gold shell by compensating for the consumption of reactants due to reduction. Each solution was continuously added dropwise to the reaction solution through the silicone tubings (Figure 1c). The volume of each droplet added through the adapter at the end of the tube was well controlled. As the solution was added dropwise to the glass vial, the color of the reaction solution containing AuNP-seeded silica NPs gradually changed from reddish-brown to navy blue. The high-throughput synthesis involving the use of the continuous infusion system yielded 10 mg of SiO_2_@Au NS, which is 50 times higher than that of the previously reported method (0.2 mg NPs), as shown in Figure 1d [16]. The colors of the two dispersions of SiO_2_@Au NS prepared by both the low-throughput (0.2 mg NPs) and high-throughput synthesis methods were similar to the naked eye.

The structures of the NPs prepared in each step of the high-throughput synthesis of SiO_2_@Au NS are shown in Figure 2a–c. The diameter of the core silica NP support was calculated as 145.5 ± 11.7 nm. The silica NPs, whose surfaces were aminated and then seeded with an excess of tetrakis(hydroxymethyl)phosphonium chloride (THPC)-stabilized AuNPs, had numerous AuNPs with a diameter of ~2.3 nm. After the seed-mediated growth of SiO_2_@Au NS by the continuous dropwise addition of both the gold precursor and ascorbic acid solutions, transmission electron microscopy (TEM) image of the resulting SiO_2_@Au NS showed closely packed AuNPs covering the silica NP surface without anisotropic aggregation. The controlled gold shell structure achieved upon the continuous dropwise addition was presumed to be due to the maintenance of the reactant concentration at a steady state, as both the gold precursor and ascorbic acid solutions were gradually consumed during the nucleation and growth of the AuNPs. After the separation of the grown SiO_2_@Au NS and individual AuNPs in the growth mixture by centrifugation, the corresponding weights were measured as shown in Appendix A. In the case of the growth condition with 4.4 mL of gold precursor solution, the weight of the grown SiO_2_@Au NS in the pellet increased from 10 mg to 33.1 mg. Meanwhile, the weight of individual AuNPs in the supernatant was measured as 8.71 mg, confirming that 20.1% of the total amount of added gold precursor was consumed. As such, a significant proportion of gold precursor was involved in the growth of the gold shell, and this is estimated to be proportional to the surface area of the AuNP-seeded silica NPs that provide the growth site.

The atomic distribution of the prepared SiO_2_@Au NS was confirmed by energy-dispersive X-ray spectroscopy (EDS) mapping analysis (Appendix A). Silica nanoparticles can be identified because the inner core exhibits an EDS signal of silicon and oxygen, indicating a silicate (SiO_x_) composition. In the layered mapping image, the gold shell located outside the silica particles can be identified.

Moreover, the UV-visible absorption spectrum of SiO_2_@Au NS showed a clear LSPR band over a wide spectral range corresponding to the closely packed gold shell structure (Figure 2d). The red shift of the LSPR bands observed in the spectrum of SiO_2_@Au NS with the closely packed structure can be explained by the plasmonic coupling between neighboring AuNPs. The wide range of LSPR frequencies across the NIR spectral region of SiO_2_@Au NS is suitable for use with a 780 nm laser with minimal interference from other optical responses and high scattering efficiency [21]. In addition, the abundant nanogaps in the gold shell can serve as hotspots that generate strong local field enhancement, which is advantageous for the use of SiO_2_@Au NS as a highly sensitive SERS substrate. The abovementioned results indicate that SiO_2_@Au NS with a hotspot structure, which imparts advantageous properties for use as an optical substrate, can be synthesized in large quantities through the continuous addition of both the gold precursor and ascorbic acid solutions by using a dual-channel syringe pump.

To investigate the effect of controlling the addition rate of the reagent solution on the formation of the gold shell, either the gold precursor or ascorbic acid solution was first added in batches, followed by the continuous dropwise addition of the other solution through an infusion pump system. The TEM images of SiO_2_@Au NS prepared under dropwise addition conditions showed that the AuNPs grew differently with various sizes and shapes on the surface of the core silica NPs (Appendix A). When the ascorbic acid solution was added continuously after the addition of the gold precursor solution, relatively large and polydisperse AuNPs were irregularly bonded to the silica NPs to form a core–island structure. Another method, in which the gold precursor solution was continuously added after the addition of the ascorbic acid solution, resulted in a relatively uniform morphology of the gold shell, whereas there was a large morphology variation in the morphology among the gold shells of the different core silica NPs. The formation of irregularly structured gold shells via batch addition is presumed to be due to the uncontrolled growth caused by the vigorous nucleation of AuNPs under extremely high reactant concentration conditions at the beginning of the reduction in the gold precursor. When only Au or both AA and Au were continuously added dropwise, the LSPR band was observed at frequencies over a broad range—from the visible to NIR region—unlike those observed after other methods. Given the abovementioned SiO_2_@Au NS structure, it can be predicted that the difference in the absorption range of the LSPR band is related to the interparticle distance between the AuNPs constituting the gold shell.

### 2.2. Control of Gold Shell Thickness

Based on the simultaneous dropwise addition method using the dual-channel syringe pump, we attempted to control the thickness of the gold shell by varying the volumes of the gold precursor and ascorbic acid solutions required for the growth of the AuNP seeds. During the growth of the AuNP seeds embedded in the silica core NPs, the total volumes of both the gold precursor and ascorbic acid solutions were adjusted from 0 mL to 4.2 mL. The structure of SiO_2_@Au NS prepared at each volume of the precursor solution was analyzed using TEM; the results are shown in Figure 3a. In the TEM images, monodispersed SiO_2_@Au NS was observed at all five volumes of the precursor solution. In the high-magnification images, SiO_2_@Au NS showed a structure in which the AuNPs were well-ordered on the surface of the core silica NPs, and the thickness of the gold shell gradually increased as the addition volume increased. For each volume of the precursor solution, the sizes of the AuNPs constituting the gold shell were measured for 30 particles and compared; the results are shown in the violin plot in Figure 3b. The size distribution and mean size of the 25th to 75th percentiles of all SiO_2_@Au NS gradually increased with an increasing additional volume of Au^3+^ solution. Based on the size of the AuNPs measured from the equivalent spherical diameter, the median AuNP diameters were calculated to be 2.3, 8.3, 11.7, 13.2, and 17.2 nm as the volume of the added precursor solution increased. The measured AuNP sizes were distributed in the interquartile ranges of 0.62, 2.45, 3.35, 4.42, and 2.42 nm.

The LSPR frequency of metal nanocomposites can vary depending on their structures despite the consistent composition of metals. In the case of nanoshell structures, the size of the core particles and the thickness of the metal shell can act as major factors affecting the LSPR characteristics [15]. The absorption spectra of SiO_2_@Au NS grown at different precursor solution volumes based on the core silica NPs of the same size are shown in Figure 3c. In the case of the particles with AuNP seeds arrayed before the growth step, the background-level absorption was similar to that observed for the core silica NPs. However, clear bands with maximum absorption wavelengths of 558, 609, 629, and 648 nm appeared as the precursor solution volume increased to 1.1, 2.2, 3.3, and 4.4 mL, respectively. Along with a redshift of the peak LSPR wavelength, an increase in absorbance was observed. At 780 nm, which is the wavelength of the laser source used for SERS analysis, the absorbance increased in proportion to the size of the AuNPs constituting the gold shell. These results demonstrate that the fine-tuning of the gold shell thickness during the seed-mediated growth process and the corresponding tuning of plasmonic properties can be achieved at an improved scale of 10 mg.

### 2.3. SiO_2_@Au NS as SERS Substrate

Based on the hotspot structure and plasmonic properties of SiO_2_@Au NS prepared using the high-throughput synthesis method, Raman spectroscopy was performed after adsorbing 4-aminothiophenol (4-ATP) as a Raman label to evaluate the performance of SiO_2_@Au NS as a SERS substrate. To demonstrate the effect of the gold shell thickness on the SERS activity, an excess of 4-ATP (10 mM) was adsorbed on each SiO_2_@Au NS with different gold shell thicknesses. The Raman spectra from five kinds of SiO_2_@Au NS with gold shell thicknesses ranging from 2.3 to 17.2 nm were compared as shown in Figure 4a. The SiO_2_@Au NS, in which the AuNP seeds of 2.3 nm size were arranged, showed no signal other than the background signal. However, as the gold shell thickness increased, the Raman signal at 1078 cm^−1^ (the major band of 4-ATP) increased proportionally. This signal enhancement was quantitatively confirmed by comparing the intensities at 1078 cm^−1^, as shown in Figure 4b. Based on that of the 2.3 nm-sized AuNP-seeded silica NP before the growth, the peak SERS intensity was calculated to be enhanced by 18, 78, 138, and 162 times at SiO_2_@Au NS with gold shell thicknesses of 8.3, 11.7, 13.2, and 17.2 nm, respectively. Comparing the absorption at 780 nm, which is the laser irradiation wavelength for SERS analysis, it is almost consistent with the trend of the SERS signal corresponding to the gold shell thickness. The relationship between the plasmonic resonance and SERS intensity matched well with those previously reported for various nanostructures [22,23]. Based on this relationship observed for SiO_2_@Au NS, it can be inferred that a stronger electromagnetic field enhancement occurred as a result of the enhanced dipole resonances of LSPR in the thicker shell. Therefore, the proposed SiO_2_@Au NS fabrication method is successful as a high-throughput synthesis method capable of controlling LSPR and SERS, which are the main characteristics of plasmonic NP-based sensors.

To optimize the SERS labeling process, the treatment concentration of 4-ATP solution was varied from 0.01 to 10 mM, and the corresponding Raman spectra were measured and compared; the results are shown in Figure 4c. In the case of SiO_2_@Au NS, with a shell thickness of 17.2 nm, all the particles treated with 4-ATP at a concentration of ≥0.05 mM showed clear SERS signals at the characteristic Raman scattering frequencies (387, 1078, and 1587 cm^−1^) of 4-ATP. In the plot comparing the peak SERS intensity at 1078 cm^−1^, a drastic signal change between the blank solution and the 0.1 mM solution was observed; however, at higher concentrations, the signal intensity did not increase, and the signal became saturated (Figure 4d). It can be assumed that the amount of 4-ATP adsorbed on the hotspots of SiO_2_@Au NS reached its upper limit at 0.1 mM.

To compare the SERS activities on the two kinds of SiO_2_@Au NS prepared by the proposed high-throughput synthesis method and the reported method, the ratio between the reactants was adjusted to be the same (mg NP/μmol Au^3+^ = 1/11) during the seed-mediated growth process. The SiO_2_@Au NS synthesized by both the high-throughput and the previously reported methods exhibited a gold shell surface closely packed with AuNPs in TEM images (Appendix A).

In relation to that prepared by the previously reported method, the SiO_2_@Au NS prepared by the high-throughput synthesis method exhibited similar shapes of LSPR bands across the entire UV-visible region, with almost the same peak absorbance wavelengths at 600 nm (Figure 5a). This is because both the nanostructures of the SiO_2_@Au NS prepared by the reported and proposed methods were similar. In the SERS spectra measured after labeling with 4-fluorothiophenol (4-FBT) under the optimized conditions, the SiO_2_@Au NS prepared by the reported and the proposed methods showed the characteristic signals corresponding to the vibrational modes (Figure 5b). When comparing the peak SERS intensity at 1074 cm^−1^, as shown in Figure 5c, the signals of almost similar intensities were observed with a coefficient of variation (CV) of 5.46%. These results suggest that SiO_2_@Au NS prepared by the proposed high-throughput synthesis method at a 50-times higher scale exhibited SERS activity comparable to that of previously reported gold nanoshells and that of the SiO_2_@Au NS prepared by the previously reported method. Therefore, the high-throughput synthesis method of SiO2@Au NSs proposed in this work can achieve improved productivity for industrial applications while maintaining tunable LSPR and SERS characteristics.

## 3. Materials and Methods

### 3.1. Chemicals

Tetraethyl orthosilicate (TEOS), THPC, gold(III) chloride, PVP (average molecular weight ≈ 10,000), APTS, AA, 4-FBT, and 4-ATP were purchased from Sigma-Aldrich (St. Louis, MO, USA). Ethanol (≥99.9%) and aqueous ammonia solution (NH_3_, 25–28%) were purchased from Daejung Chemicals and Metals (Siheung, Republic of Korea). Deionized water was obtained using an AquaMAX Ultra 370 water purification system (Younglin Instruments, Anyang, Republic of Korea).

### 3.2. Characterization

TEM sample was prepared by drop casting the dispersion of NPs on the formvar-coated 200 mesh copper grids. TEM analysis was performed using a JEM-1010 instrument (JEOL, Tokyo, Japan) operated at an acceleration voltage of 80 kV. The UV-Vis absorption spectra were measured using a single-beam-type UV-Vis spectrophotometer (U-5100, HITACHI, Tokyo, Japan). The sizes of the AuNPs were calculated from at least 30 particles in the TEM images using the ImageJ 1.52a software (National Institutes of Health, Bethesda, MD, USA).

### 3.3. Synthesis of AuNP Seeds

The AuNPs were prepared using THPC as a reducing agent and stabilizer [24]. First, the NaOH solution (0.2 M, 1.5 mL) was diluted by mixing with deionized water (47.5 mL) in a 100 mL round bottom flask. Next, the THPC solution (80%, 12 μL) and gold(III) chloride solution (50 mM, 1 mL) were added sequentially. In alkaline conditions, THPC generates both formaldehyde and hydrogen which can act as an effective reducing species [25]. Due to the formation of ultra-small AuNP seeds through this mechanism, the solution turns dark brown as soon as the Au^3+^ precursor solution is added. The mixture was stirred vigorously for 1 h. The prepared colloidal AuNPs were stored in a refrigerator at 4 °C.

### 3.4. High-Throughput Synthesis of SiO_2_@Au NS

SiO_2_@Au NS was prepared using silica NP support. First, silica NPs were prepared following the Stӧber process [26]. Briefly, TEOS (1.6 mL) was mixed with ethanol (40 mL) in a round bottom flask, and the NH_3_ solution (3–5.5 mL) was added to obtain silica NPs of various sizes. The reaction mixture was stirred vigorously for the first 1 h at 60 °C, followed by stirring for 19 h at room temperature. The solution was washed several times with ethanol via centrifugation at 9000× *g*. Next, the resulting silica NPs (15 mg) were modified with amino groups by stirring with APTS (15.5 μL) and NH_3_ solution (10 μL) in a vortex mixer. After stirring for 15 h, the mixture was washed several times with ethanol via centrifugation. The aminated silica NPs (2 mg) were then mixed with the AuNP solution (10 mL), and the mixture was stirred overnight to prepare AuNP-seeded silica NPs. After washing with deionized water, the resulting dark brown pellets were dispersed in ethanol. Using the as-prepared AuNP-seeded silica NPs, SiO_2_@Au NS was synthesized using a newly optimized seed-mediated growth process. The AuNP-seeded silica NPs (10 mg) were dispersed in the PVP solution (100 mL, 1 mg/mL PVP) and transferred to a glass vial. The vial was then combined with an infusion pump system loaded with an aqueous solution of gold(III) chloride (50 mM) and ascorbic acid (100 mM). While the dispersion was being stirred, each solution was added simultaneously at a fixed flow rate (36.67 μL/min) for 0.5, 1, 1.5, and 2 h to adjust the final volume of each solution to 1.1, 2.2, 3.3, and 4.4 mL. The resulting mixture was sequentially washed with deionized water and ethanol at 3000 g. The pellet was then re-dispersed in ethanol (10 mL) and stored in a refrigerator.

### 3.5. SERS Measurement

For SERS analysis, SiO_2_@Au NS (0.2 mg) was incubated with an ethanolic solution (0.5 mL) of each Raman label (4-ATP or 4-FBT). After incubation for 1 h in a vortex mixer, the resulting mixture was separated via centrifugation at 15,000× *g* for 10 min. The separated pellet was washed three times and redispersed in ethanol. Raman spectroscopy was performed using a DXR3 Raman microscope (Thermo Fischer Scientific, Waltham, MA, USA) coupled with a 780 nm high-power laser. The laser was focused in the middle of a glass capillary filled with colloidal SiO_2_@Au NS (1 mg/mL). The Raman spectra were collected at an exposure time of 16 s and a laser power of 150 mW. All data represent the average of three measurements and were subtracted from the background Raman signal of each SiO_2_@Au NS without Raman labels.

## 4. Conclusions

In summary, this study proposed a high-throughput synthesis method for SiO_2_@Au NS at a scale 50 times higher than that of previously reported methods. Monodispersed SiO_2_@Au NS with closely packed structures was prepared without the uncontrolled aggregation of AuNPs by using a double-channel syringe pump during the addition of the precursor solution. SiO_2_@Au NS prepared by the high-throughput synthesis method showed strong SERS signals for the adsorbed Raman labels, similar to those observed for the sample synthesized by a previously reported method. Due to the easily controllable structures and SERS activity of SiO_2_@Au NS, the proposed high-throughput synthesis method can be used for fabricating tunable plasmonic sensors for industrial applications. The limitations of the proposed method include (1) difficulty in achieving a gold shell thicker than 17.2 nm by adding the precursor solution dropwise above the reported volume and (2) challenging separation of large individual AuNPs derived from excess gold precursors. By overcoming these limitations, the investigation of the correlation between absorption and SERS scattering activity at high thicknesses, particularly in the range of tens to hundreds of nanometers, will be possible.

## Figures and Tables

**Figure 1 ijms-25-01649-f001:**
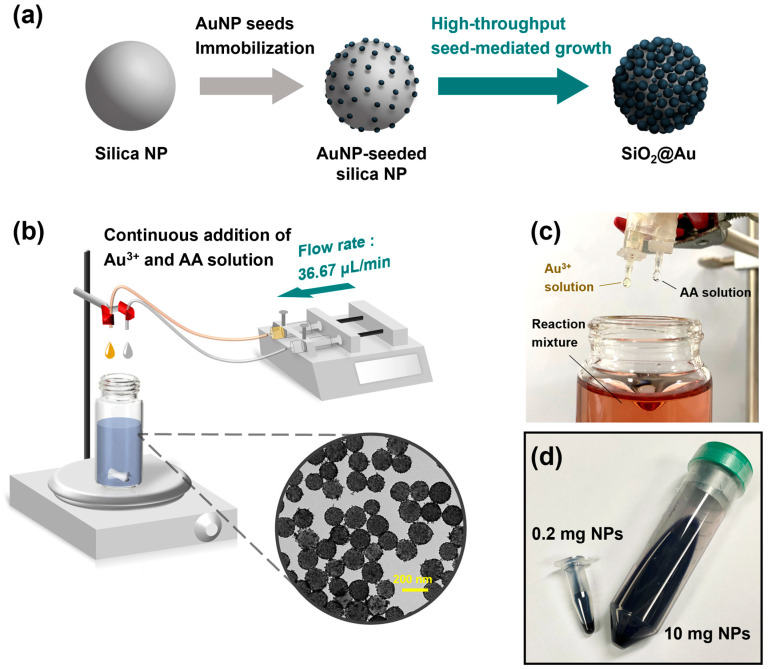
Schematic illustrations of the high-throughput synthesis of SiO_2_@Au NS. (**a**) Two-step SiO_2_@Au NS fabrication. (**b**) Optimized continuous addition system for the AuNP-seed-mediated growth step equipped with a dual-channel syringe pump. Au^3+^ and AA solutions were loaded in 15 mL syringes and infused through silicone tubings at a flow rate of 36.67 μL/min. Inset: transmission electron microscopy image showing the morphology of the monodispersed SiO_2_@Au NS. Photographic images of (**c**) the reaction mixture during the continuous addition of precursor solutions and (**d**) the resulting colloidal SiO_2_@Au NS prepared at scales of 0.2 mg and 10 mg.

**Figure 2 ijms-25-01649-f002:**
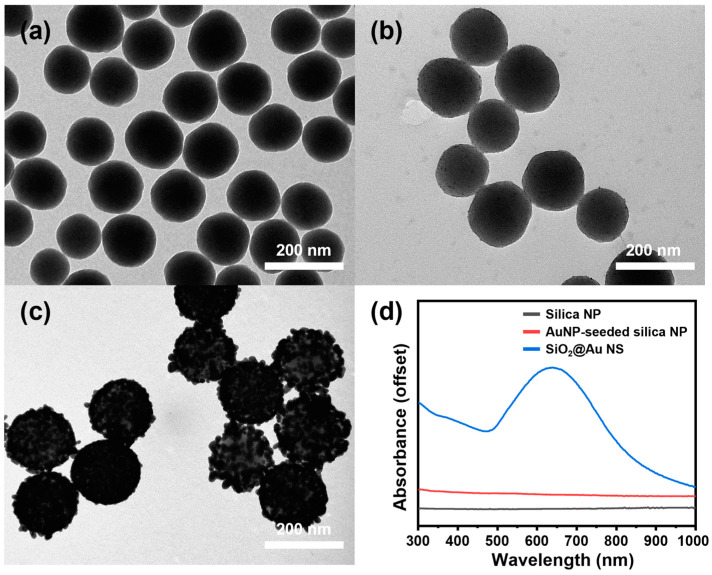
TEM images of the (**a**) silica NPs, (**b**) AuNP-seeded silica NPs, (**c**) SiO_2_@Au NS, and their corresponding (**d**) UV-visible absorption spectra with offset.

**Figure 3 ijms-25-01649-f003:**
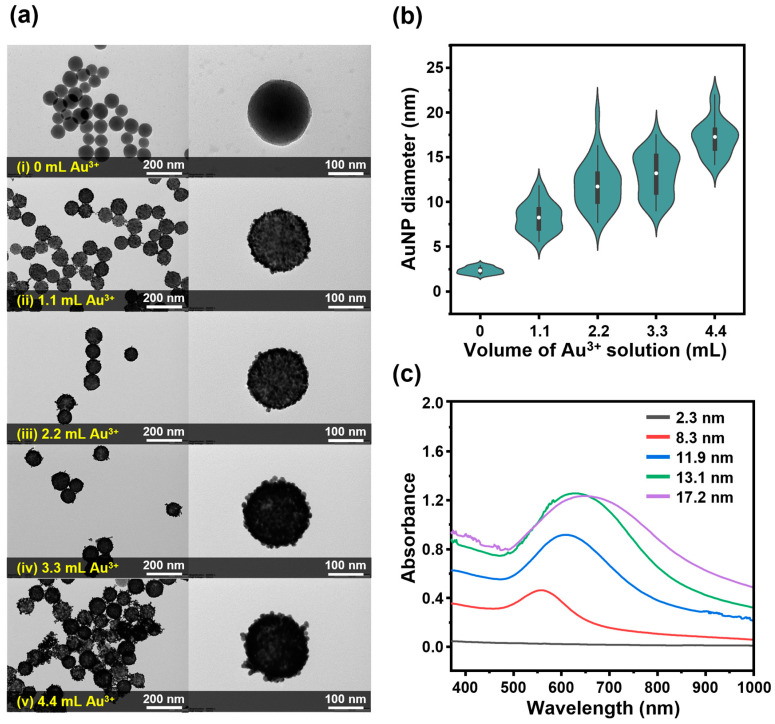
(**a**) TEM images of SiO_2_@Au NS synthesized at different volumes of both the Au^3+^ precursor and AA solutions: (i) 0 mL, (ii) 1.1 mL, (iii) 2.2 mL, (iv) 3.3 mL, and (v) 4.4 mL at low-magnification (**left**) and high magnification (**right**). (**b**) Violin plot of the equivalent spherical AuNP diameter measured for at least 30 particles in the TEM images. Black line shows the 25–75th percentile, while the open circle corresponds to the mean. (**c**) UV-visible absorption spectra of each SiO_2_@Au NS with the median-sized AuNPs.

**Figure 4 ijms-25-01649-f004:**
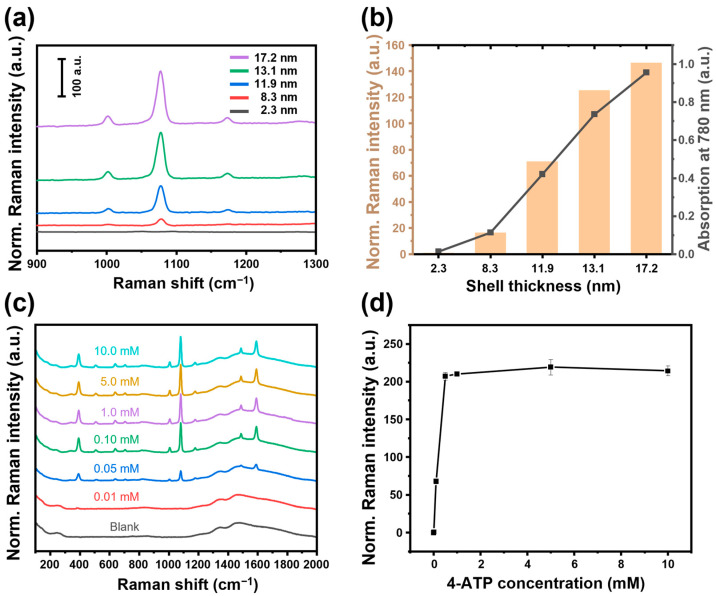
(**a**) SERS spectra of SiO_2_@Au NS with different median AuNP sizes and (**b**) peak SERS intensities with the peak absorbance at 780 nm. (**c**) SERS spectra and (**d**) peak SERS intensities of SiO_2_@Au NS incubated with different concentrations of 4-ATP.

**Figure 5 ijms-25-01649-f005:**
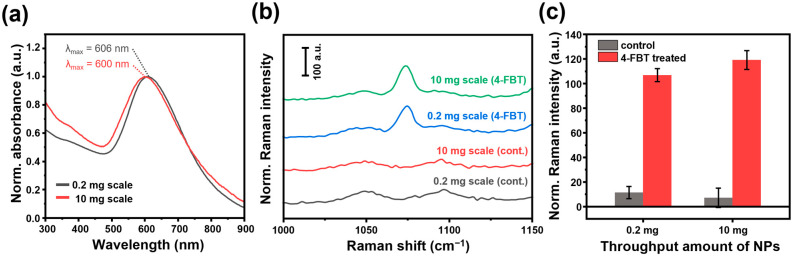
LSPR and SERS characteristics of SiO_2_@Au NS prepared by the reported method (0.2 mg) and the proposed high-throughput (10 mg) synthesis method. (**a**) Normalized absorption spectra with the maximum LSPR wavelength. (**b**) SERS spectra and (**c**) peak SERS intensity of SiO_2_@Au NS treated with blank (grey columns) and 4-FBT solution (red columns).

## Data Availability

The data presented in this study are available upon request from the corresponding author.

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
