# Peer review of "High-Throughput Synthesis of Nanogap-Rich Gold Nanoshells Using Dual-Channel Infusion System"

_ijms, 2024, doi:10.3390/ijms25031649_

Round 1
Reviewer 1 Report
Comments and Suggestions for Authors
Dear Editor/Authors,
High-throughput SERS active gold nanostructures on silica nanoparticles was developed using a continuous system. The paper is well prepared, and all the figures are quite clear. Production of uniform and high amounts of SERS active gold nanostructures are critical in order to convey the nanostructures into sensing applications. The developed method is quite unique and a new concept that can attract the attention of the researchers working in the field. Even though utilization of microfluidic system is not new in the synthesis of metallic nanoparticles (ACS nano 7.5 (2013): 4135-4150), its application in gold nano-shell on silica support is a new application. Besides, the paper used covalent linkage of the seeds to the silica supports, which can enhance uniformity. The findings were discussed very well in the light of literature and all the mechanisms were slightly discussed as well.
However, there should be discussion for the gold nanoparticles or even clusters could form during ascorbic acid and Au3+ ion mixing process. That section needs to be discussed; for example, was any study performed to understand which percent of Au3+ ions turned into gold nanoparticles that were not accumulated on the silica support?
Besides, full names must be given before the abbreviations (such as 4-ATP) at their first usage?
Kind Regards,
Reviewer 2 Report
Comments and Suggestions for Authors
Kim et al. present in this manuscript the synthesis of nanogap-rich gold nanoshells based on a core silica support and the investigation of their plasmonic characteristics. In general, the manuscript is carefully written, and the results deserve publication. The problematic issue is, however, the identification of chemical species; authors should augment the manuscript with clarifying the identity of chemical species.
Comments:
--- line 85: “ammonium hydroxide” does not exist (outdated knowledge written in old textbooks). Please replace it with “aqueous ammonia solution (NH3, 25-28%)”.
--- Replace NH4OH with NH3 throughout the manuscript (see above).
--- lines 96-100 (2.3. Synthesis of AuNP seeds): at this stage only a gold(III) complex forms, not AuNPs. There is no reducing agent in the mixture. Please check the text and modify accordingly.
--- lines 101-122 (2.4. High-throughput synthesis of SiO2@Au NS): please provide the correct name of compounds or materials at each stage. Ascorbic acid is the reducing agent and AuNPs can form after adding the ascorbic acid.
--- Figure 1: 1a: “AuNP-seeded silica NP” should be “Au(3+)-seeded silica NP”. Please provide proof if you believe that you have AuNPs at this stage, e.g. by using powder X-ray diffraction.
--- In general, direct evidence of metallic Au NPs or layers should be provided, by using e.g. PXRD. PXRD could be used also to calculate the average NP size. In addition, identifying the SiO2 core is also necessary.
Comments on the Quality of English LanguageMinor editing of English is required.
Reviewer 3 Report
Comments and Suggestions for Authors
Very nice paper reporting on a very important subject. Authors report a high-throughput synthesis method for the synthesis of gold nanoshells at a much higher scale (50 times more( than method from previous literature. This is a very important achievement, knowing how difficult it is to upscale methods and still achieve good results. Moreover, the paper is well written, very clear, very detailed, with several illustrations, so the results can be easily reproduced, if wanted. Also the results are very good.
I just have one question. Authors refer their material as SiO2@Au. By looking at Figure 1, shouldn't it be the other way around? That is Au@SiO2? As described, gold is deposited on SiO2, so usually, when that happens, the way I have been seen, that seems more common to me is "material deposited"@"support". I do not mean to say that what authors wrote is wrong, I just want to suggest a clarification, as this was the only thing I had to point out to a very good paper.
